# Geographic Distribution of Common Vampire Bat *Desmodus rotundus* (Chiroptera: Phyllostomidae) Shelters: Implications for the Spread of Rabies Virus to Cattle in Southeastern Brazil

**DOI:** 10.3390/pathogens11080942

**Published:** 2022-08-19

**Authors:** Karine B. Mantovan, Benedito D. Menozzi, Lais M. Paiz, Anaiá P. Sevá, Paulo E. Brandão, Helio Langoni

**Affiliations:** 1Departamento de Produção Animal e Medicina Veterinária Preventiva, Faculdade de Medicina Veterinária e Zootecnia, Universidade Estadual Paulista “Julio de Mesquita Filho”, Botucatu 18618-681, São Paulo, Brazil; 2Departamento de Saúde Coletiva, Faculdade de Ciências Médicas, Universidade de Campinas, Campinas 13083-887, São Paulo, Brazil; 3Departamento de Ciências Agrárias e Ambientais, Universidade Estadual de Santa Cruz, Ilhéus 45662-900, Bahia, Brazil; 4Departamento Medicina Veterinária Preventiva e Saúde Animal, Faculdade de Medicina Veterinária e Zootecnia, Universidade de São Paulo, São Paulo 05508-270, São Paulo, Brazil

**Keywords:** vampire bat, cattle, shelters, rabies

## Abstract

*Desmodus rotundus* bats show a complex social structure and developed adaptive characteristics, considered key features of a pathogen disseminator, such as the rabies virus, among bats and other mammals, including cattle and humans. Our aim was to understand the correlation between the environment and the ecological features of these bats in bovine rabies outbreaks. Geostatistical analyses were performed, covering 104 cattle positives for rabies, between 2016 and 2018, in 25 municipalities, in addition to the characteristics of *D. rotundus* colonies mapped during this period in the state of São Paulo, Brazil. Data from the shelters showed that 86.15% were artificial, mainly abandoned houses (36.10%) and manholes (23.87%), in addition to demonstrating a correlation between these shelters and a higher concentration of bovine rabies cases. Due to their adaptive capacity, these bats choose shelters close to the food source, such as livestock. In Brazil, *D. rotundus* is the main transmitter of rabies and the cause of outbreaks in cattle and deaths in humans, considering the advance of humans in previously preserved ecosystems. There seems to be a correlation between the impact of anthropic changes on the environment, mainly for the expansion of pasture for cattle and the outbreaks of bovine rabies in this area.

## 1. Introduction

Bats (Order *Chiroptera*) are considered the main transmitters of emerging viral zoonoses due to their large capacity to act as a natural reservoir for several species of pathogens [1,2], such as the rabies virus, in comparison with other mammals.

Rabies has the highest fatality rate (100%) among all infectious diseases currently described [3] and is considered one of the world’s main neglected tropical diseases [4].

In Latin America, after controlling canine rabies through mass vaccination campaigns, the hematophagous bat *Desmodus rotundus* became the main transmitter of rabies virus to humans and farm animals [5,6] since these herbivores are the main feed source of this bat species. It is estimated that rabies infection in bats generates economic losses of US$30 million annually throughout the region [7,8].

The Brazilian territory covers extensive latitude and longitude and *D. rotundus* is present throughout this area [9], mainly due to its high adaptive capacity to a variety of habitats [2]. To achieve this adaptation, the species makes use of natural shelters, such as tree trunks and caves, or artificial shelters resulting from environmental changes of anthropogenic origin [10].

*D. rotundus* has a complex social structure and characteristics linked to altruism and social cohesion [11,12], which, in the context of this species, consists of licking to stimulate blood regurgitation, a factor considered one of the main dynamics of rabies virus transmission among bats [13]. However, the spatial dispersion of infected bats is considered slow since the virus spreads from colony to colony due to the species’ tendency to visit nearby colonies, often infecting new groups [14,15].

According to the Ministry of Agriculture, Livestock and Supply [16], in Brazil, rabies control in cattle should be based on epidemiological surveillance, herd vaccination, selective control of *D. rotundus* colonies and health education. However, based on this perspective, the methods of selective control of bats are contradictory because, rather than controlling the species, they have been associated with viral dissemination [17,18].

Given the notorious importance of *D. rotundus* in the epidemiological cycle and spatial dispersion of the rabies virus in bats and cattle, the purpose of this study was to better understand the bioecological factors associated with virus circulation in different municipalities in the state of São Paulo, Brazil. More specifically, we characterized the types of bat shelters in the region, together with their respective frequencies, and determined the spatial and temporal distribution of cases of rabies in cattle to spatially associate rabies cases in cattle with bat colonies and different types of bat shelters.

## 2. Results

The research included 1553 *D. rotundus* shelters mapped in the study area, and the classifications and categories are presented in Table 1.

Among the artificial shelters, the main construction used was abandoned houses, 36.10% (483/1338). See in Appendix A. and culverts 23.47% (314/1338). Among the natural shelters, the majority were caves (80.86%; 169/209) and hollow trees (11.48%; 24/209). See in Appendix A.

When observing the areas with the highest concentrations of *D. rotundas* shelters (Figure 1A) and cases of rabies in cattle (Figure 1B), we analyzed the overlap between areas of higher density in the same region, the central-south of the state, in the neighboring municipalities of Itatinga, Pardinho and Bofete.

Figure 2 shows the location of active *D. rotundus* shelters in relation to the road system, indicating the main highways, BR-304 (Rodovia Castelo Branco), SP-300 (Rodovia Marechal Rondon), SP-147 (Rodovia Lazaro Cordeiro de Campos), secondary roads and their relation to the density of rabies cases in cattle. Many shelters consisted of colonized anthropogenic structures along highways.

Figure 3 illustrates the range of areas used by bats surrounding different types of shelters and the overlap with rabies cases in cattle. Cases were significantly (*p* < 0.01) more present in areas in the range of *D. rotundus* single male shelters (92.3%) and harems (97.1%). Rabies cases in cattle in the range of overnight digestive and empty shelters were as low as 37.5% and 2.9%, respectively.

Significant differences in spatial dispersion were observed between cases for the 3-month periods t1–t2 (April to June 2016) and t2–t3 (July to September 2016) and between each pair of 3-month periods from t3–t4 to t7–t8 (October 2016 to December 2017) (Figure 4). However, dispersion between t1–t2 was very high with respect to period t2–t3, in which cases appeared stable.

## 3. Discussion

The already known adaptive capacity of *D. rotundus*, particularly in changes of habitat [6,19,20], can be observed in our study in the 86% (1338/1553) shelters found to be characterized as artificial, mainly abandoned houses and culverts (water drainage structures on highways) (Figure 2). In this case, the extensive road system in the region contributes to these numbers, which allows the formation of shelters in these facilities, among which culverts correspond to 20% [10,17].

Anthropogenic action is widely known as the main factor in behavioral changes in wild animal species [19], especially in those species whose shelter is related to feeding and reproduction, such as *D. rotundus* [3,8].

Studies in Brazil [19,21] and Argentina [20] do not seem to indicate a preference for the types of shelters during the occurrence of rabies outbreaks in production animals; both natural and artificial shelters (Figure 2) can be considered risk factors for rabies cases on a local scale [8].

The region studied is one of extensive livestock production, with a high density of cattle, a factor favorable to *D. rotundus*, and it is possible to verify that most of the rabies cases in cattle overlapped with the higher density of shelters of *D. rotundus* (Figure 1).

According to Streicker et al. (2012) [14], colony size and concentration are related to cattle density, and it seems probable that the combination of both is associated with the incidence of rabies [21], indicating that definitive colonies are located in areas close to the food source, the cattle.

As the abundance of vampire bats and shelters increases, so does the number of connections between them [18] and the contact between healthy and infected bats, which could explain the continuous circulation of the rabies virus in this population [14,22].

The factors favorable to sheltering *D. rotundus* previously mentioned by Rocha et al. [19], Delpietro et al. [20] and Braga et al. [23] refer to extensive livestock production, which in this region corresponds to more than 820,000 cattle [24], and the location of natural and artificial shelters, a total of 1553 in the region. Factors for vulnerability that assist in the dispersion of the species to new areas are based on deforestation for growing pasture, the movement of cattle (withdrawal/introduction) and new constructions. These conditions may help to understand the dynamics of rabies virus dispersion in the study area [10,23].

Significant differences (*p* < 0.005) in spatial dispersion were observed between rabies cases in cattle in the 3-month periods between 2016 and 2017, suggesting that the virus migrated from one region to another (Figure 4). One of the factors, described as “environmental vulnerability,” that can influence the migration of *D. rotundus* and the consequent spread of the virus to new areas is the movement of cattle herds [14,23]. In our research, some municipalities recorded the withdrawal/introduction of at least 10% of the herd size, about 90,524 animals [24].

Thus, the abrupt movements of the herd influenced the dynamics of the shelters.

Field observations have already shown that the abandonment of shelters by species *D. rotundus* occurs when cattle move away [22], indicating that the bat also chooses its shelter due to the proximity of its food source. Furthermore, rabies outbreaks transmitted by *D. rotundus* are already known to move slowly, at an average rate of 40 km/year [1,15].

In general, foraging by *D. rotundus* depends on the diversity and dynamics of the environment in which they live [25]. In this context, certain topographic and geographic factors are associated with hematophagous bat attacks on cattle due to their ability to adapt physiologically to changes in environmental conditions [6], such as the migration patterns of outbreaks of rabies infection [15,26].

The classification of *D. rotundus* shelters according to their social uses indicated two main types: harems (42.63%; 662/1553), containing adult and young females and a few males, and single males (youth and adults) (44.88%; 697/1553), generally located close to harem shelters, were those most associated with rabies cases in cattle (*p* < 0.01). This relation is explained by the proximity of both types of shelters (Figure 3), which overlap with cattle production areas. The risk of rabies infection will always be higher in areas neighboring bat shelters that are connected [18] since the movement of infected bats between shelters is extremely important for viral maintenance [3,22].

Only 9.34% of shelters were categorized as overnight spaces [11,27] used to complete digestion [28], to minimize energy loss during foraging, to protect against predators, and for social interactions [21,22]. Since the use of these intermediate shelters was not significant, this indicates that the definitive colonies are in areas close to the food source, in this case, the cattle, which is in agreement with that reported by Rocha et al. (2020) [19]. Thus, there seems to be a relationship between the location of permanent shelters close to the food source since this requires a lower energy demand from bats foraging for food [28].

The investigation of rabies transmitted by *D. rotundus* is important for identifying the presence of virus circulation in a given area [26] although no human case has been reported in the study area. In practice, the control of zoonosis depends on the identification of the so-called event sentinel, the first death, which normally occurs in cattle and later in humans [27,28]. Therefore, the authorities responsible for health and surveillance actions should be attentive to the reported locations in order to minimize the spread of the disease to new areas [17,29], and areas of high livestock density where rabies is commonly silent should continue to be investigated [30].

## 4. Materials and Methods

### 4.1. Animals Evaluated

In the state of São Paulo, 104 samples from the central nervous system of cattle infected by the rabies virus were collected in 25 municipalities from 2016 to 2018 (Figure 5). Diagnoses were performed to identify infection by rabies virus using the techniques recommended by the World Health Organization, direct immunofluorescence reaction and the mouse inoculation test [31], at the Zoonosis Diagnostic Service of the School of Veterinary Medicine and Animal Science (FMVZ), São Paulo State University (UNESP), Botucatu, São Paulo, Brazil.

Geographic coordinates were obtained whenever samples were collected, contained in the submission form, for posterior spatial analysis.

The project was approved by the Animal Ethics Committee of the FMVZ-UNESP, Botucatu, under protocol no. CEUA 0066/2018.

### 4.2. Bat Shelters

Data were evaluated from all bat shelters located in areas covered by the nine Agricultural Defense Offices (*Escritórios de Defesa Agropecuária*, EDA), responsible for the 25 municipalities where cases of rabies in cattle occurred. The EDA is the state body responsible for implementing the Herbivore Rabies Control Program, which is administrated by the State Department of Agriculture of the Government of São Paulo [32].

*D. rotundus* shelters were registered during active surveillance activities from 2015 to 2018 and were visited annually by the EDA team. On these occasions, the type of roost and its occupation were recorded, classified into four types and additional categories: (1) use: (a) harem, occupied mainly by females, young bats and a dominant male; (b) single, occupied by young males; (c) overnight, used as a temporary resting stop during foraging, digestion and social interaction; and (d) empty; (2) type: (a) natural; (b) artificial; and (c) not informed; (3) status: (a) active; (b) destroyed; and (4) others, which include shelters (a) destroyed; (b) deactivated; and (c) not informed [16].

The characterization of the shelters was determined by visiting the sites, which was conducted throughout 2018.

### 4.3. Spatial Analysis

Properties where cattle presented rabies infection were georeferenced using Google Maps (https://www.google.com/maps). Georeferencing of *D. rotundus* shelters in the state of São Paulo was obtained using a GPS device by the Agricultural Defense Coordination.

The kernel density estimate was generated to identify the spatial density of *D. rotundus* shelters and rabies cases in cattle using a 10 km radius, the quartic model, and full-scale output value as parameters.

Areas of influence of bats in shelters: A buffer was created around each bat shelter, with a radius of 10 km, corresponding to the area used by bats surrounding the shelter [19]. This area was the bat’s range to compare how it overlapped with rabies cases in cattle. Comparisons between each type of area and case occurrence were performed by the chi-square test, which was considered a significance value of *p* < 0.05.

Cases of rabies in cattle were clustered every 3 months; thus, there were eleven 3-month periods (p1–p11) over the period evaluated: April 2016 to December 2018. The spatiotemporal distances between the cases of each subsequent period were calculated, where t is the time distance between p_i_–p_i−1_, for a total of ten time periods (t1–t10). Subsequently, comparisons were generated between pairs of subsequent time periods: t_i_ with respect to t_i−1_. For each time pair, data normality was determined using the Shapiro–Wilk test. Since the data did not show normal distribution, the non-parametric Wilcoxon test was used to assess the difference between the distances in the cases between subsequent time pairs (e.g., t1–t2 distance pair compared with t2–t3 distance pair, and so on). A value of *p* < 0.05 was considered significant. This analysis was performed to identify whether there were differences in dispersion from one period to another.

The shelters, the rabies cases in cattle, the kernel analysis and the areas of influence of the bat shelters were mapped and determined using QGis software (version 3.18.1) (GNU general public license). https://www.qgis.org/en/site/ (accessed on 20 October 2021).

## 5. Conclusions

The most relevant factors for the episodes of rabies outbreaks in cattle observed in this study were the adaptive capacity of the bat species *D. rotundus* to artificial shelters, corresponding to more than 86% of the shelters studied, and the proximity to the food source, the cattle. These factors highlight the impact of anthropogenic environmental changes and the importance of the surveillance of virus circulation. The risk of interactions between bats, cattle and humans must be considered now more than ever.

## Figures and Tables

**Figure 1 pathogens-11-00942-f001:**
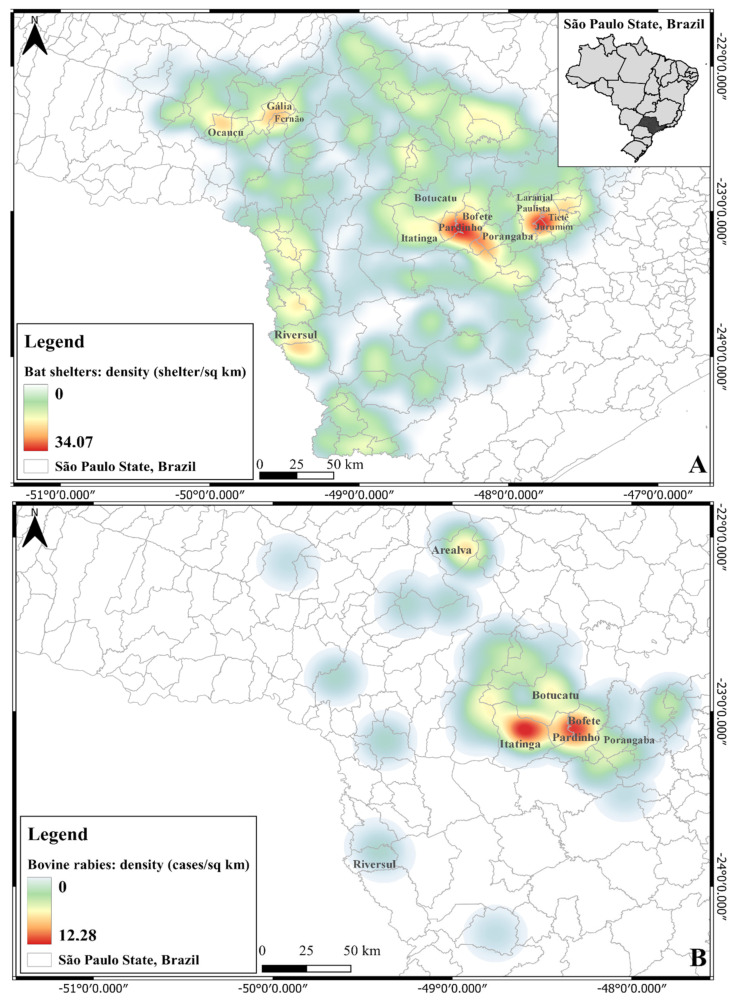
Density of *D. rotundus* shelters in 2018 (**A**) and rabies in cattle between 2016 and 2018 (**B**) in municipalities in southeastern Brazil.

**Figure 2 pathogens-11-00942-f002:**
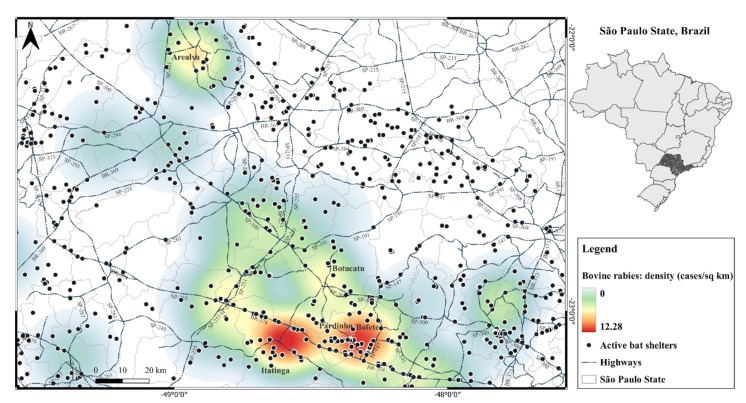
Location of *D. rotundus* shelters along the road system and density of rabies in cattle between 2016 and 2018 in southeastern Brazil.

**Figure 3 pathogens-11-00942-f003:**
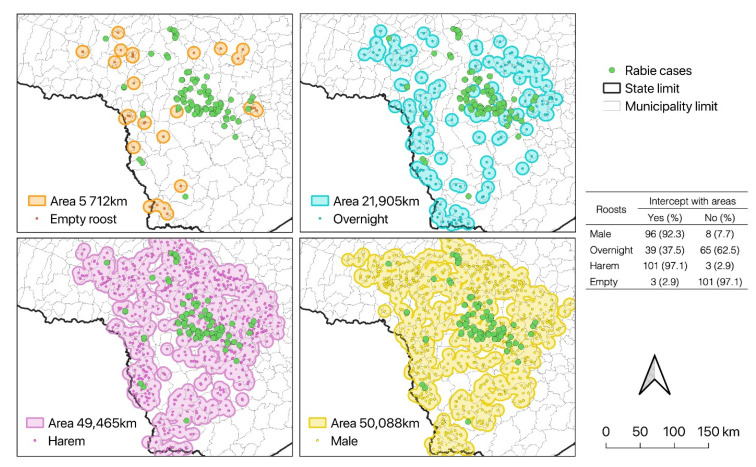
Geographical area used by *D. rotundus*: shelters found and location of rabies cases in cattle in southeastern Brazil. Cases of rabies in cattle, which are intercepted or not in areas covered by shelters.

**Figure 4 pathogens-11-00942-f004:**
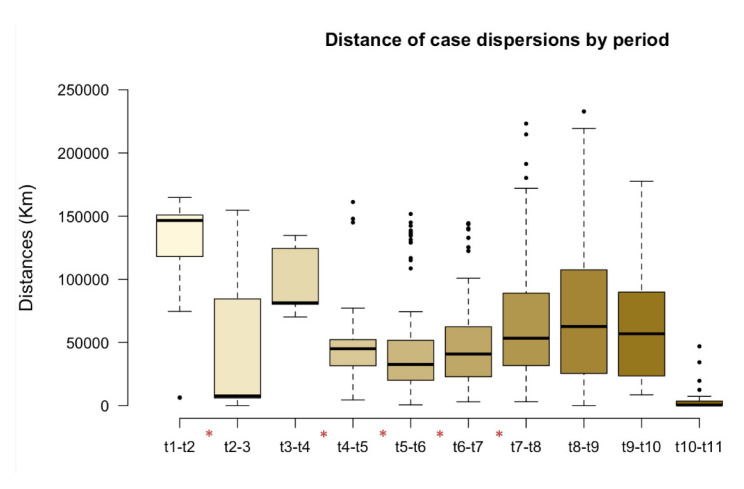
Dispersion distance by quarterly period (t) of rabies cases in cattle. t1 (April–June/2016), t2 (July–September/2016), t3 (October–December/2016), t4 (January–March/2017), t5 (April–June/2017), t6 (July–September/2017), t7 (October–December/2017), t8 (January–March/2018), t9 (April–June/2018), t10 (July–September/2018), t11 (October–December/2018). The intervals between periods containing asterisks represent significant differences between the distances of cases (*) = *p* < 0.05.

**Figure 5 pathogens-11-00942-f005:**
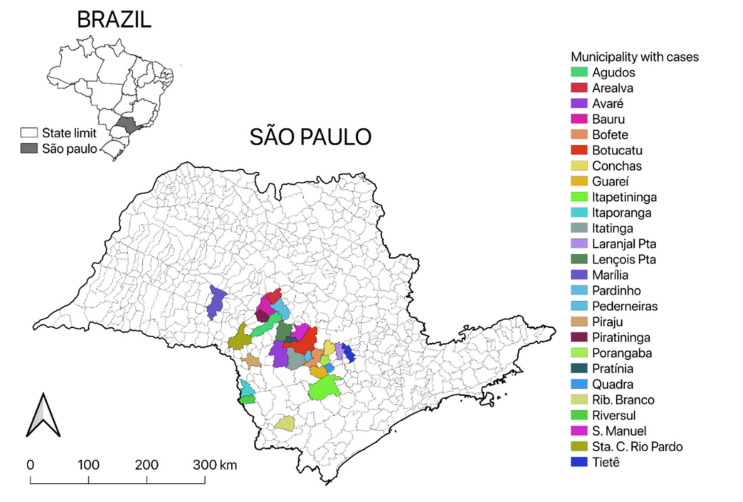
Municipalities with bovine rabies cases between 2016 and 2018 in southeastern Brazil.

**Table 1 pathogens-11-00942-t001:** *D. rotundus* shelters listed and classified into categories in southeastern Brazil between 2016 and 2018.

Shelter	Caracterization	(n) (%)	Total
Type	Artificial	1338 (86.15%)	1553
	Natural	209 (13.46)	
	Not informed	6 (0.39%)	
Situation	Active	1318 (84.87%)	1553
	Destroyed	235 (15.13%)	
Use	Single male	697 (44.88%)	1553
	Harem	662 (42.63%)	
	Overnight	145 (9.34%)	
	Empty	36 (2.32%)	
	Others	13 (0.83%)	

Legend (n): number of shelters.

## Data Availability

Data are contained within the article.

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
