# Peer review of "Geographic Distribution of Common Vampire Bat Desmodus rotundus (Chiroptera: Phyllostomidae) Shelters: Implications for the Spread of Rabies Virus to Cattle in Southeastern Brazil"

_pathogens, 2022, doi:10.3390/pathogens11080942_

Round 1

Reviewer 1 Report

This is a very interesting research report on rabies in domestic animals and humans caused by D. rodundus in the state of São Paulo, Brazil, which attempts to clarify the mode of prevalence by comparing geographical environmental factors with the ecology and habitat characteristics of bats.

However, the paper does not adequately evaluate and discuss the significance of the paper because it does not cite important epidemiological information necessary to prove its discussion of the cause of rabies in cattle and humans caused by bats, and the vocabulary used in the discussion is not well defined. The paper could be more valuable by restructuring it with the necessary information.

The following is a list of areas for consideration.

In the abstract and conclusion of this paper, the author states that social structure, well-developed adaptive characteristics, types of shelters and anthropic changes are important as causes of bats causing rabies in cattle. However, the figures and tables in the paper do not provide sufficient evidence to draw conclusions. The reason for this is that the paper does not define these terms in a way that accurately reflects the information from the figures and tables.

In Figure 2, the author compares the rabies frequency in cattle with the shelter distribution in bats, but the shelter distribution on the route appears to be evenly distributed on the map.

In addition, the number of healthy animals in the host population is important for epidemiological consideration of rabies frequency, but this paper makes comparative judgments by showing only the number of individuals who tested positive for rabies, and does not accurately consider background environmental factors, host distribution and density, and other factors.

From the figures and tables presented by the author, it can be inferred that what characterizes the areas where rabies frequently occurs in cattle is not the type and distribution of shelters used by bats, but rather the number of farms and the number and distribution density of cattle raised in the areas where rabies occurs, and the influence of human living environment and density related to these factors.

Therefore, it is recommend that the author reconsider, examine, and select the parameters necessary to compare the causes in order to add accurate considerations, and come up with an argument that makes the consequences clear.

The author considers the change in t1-t2 to be significant in Figure 4, but it needs to be shown whether this feature is repeated every year. It is also necessary to consider whether anthropogenic intervention in the bat environment during the period when the study was initiated and started may have had an effect. If the changes are repeated every year, it would be very interesting to see if there is a relationship with the timing of bat reproduction.

There is a section of the text where the argument was unclear, so I will excerpt it.

(1) […. This relation is explained by the proximity of both types of shelter (Figure 3), which overlap with cattle production areas. ….], (2) [….. One characteristic of the vulnerability of the environment, which may have assisted the migration of D. rotundus and the spread of the virus to new areas was the movement of cattle herds….], (3) […. the abandonment of shelters by D. rotundus occurred when cattle were moved 1.8 km further away, ….. In addition, it has been established that rabies outbreaks transmitted by D. rotundus move slowly, at an average rate of 40 km/year.]

Reviewer 2 Report

This is an interesting study which considers the geographic distribution of vampire bat Desmudus rotundus (Chiroptera: Phyllostomidae) shelters and  the implications for the spread of rabies virus to cattle in southeastern Brazil.  It was observed that 86.15% of the bat shelters were artificial, including abandoned houses (36.10%) and manholes (23.87%), could there be some photographs of these shelters provided ? What is the typical habitat range of these bats ? Is 10 Km a reasonable proxy ? Although it was concluded that there is a correlation between the location of artificial shelters and a higher concentration of bovine rabies cases (which is interesting) there seems to be little consideration of other factors such as cattle population density and vaccination status. Can this be added ?.Do you have any data about feeding practices of the bats and cattle biting rates ? Are there Ro predictions for rabies spread in different regions of the study area ? Does anthropogenic activity disperse bats and/or contribute to enhanced human cases of rabies ? The methodology indicates that the WHO guidelines for rabies diagnosis were used, are these the same as those recommended for cattle by the World Organization for Animal Health ? The manuscript is interesting but hard to follow in places, moderate revisions for language are required , for example -ecological characteristics vs aspects.

Round 2

Reviewer 1 Report

The author seems to have made more suggestive statements about the ecology of D. rotundus involved in the rabies epidemic and its relationship with its living environment (shalter, topography, and cattle).

Hopefully, Table 1 and Figure 3 should be devised to clarify the causal relationship between Shelter's categories and bat life forms and reproductive activities. By this, can the author show the tendency of the relationship between Type, Situation, and Use?

For example, a table or diagram that hierarchically compares these items, such as comparing the proportions of each item of Situation or Use for the type of shelter, is created. As a result, wouldn't the dynamics of rabies virus dispersion in the study that the author describes in his discussion be discussed in more detail?

I hope that my suggestions will provide you with an opportunity to increase the value of your thesis before you finish proofreading it.

Author Response

We appreciate all the criticisms, suggestions and considerations intended to improve our thesis.

Yours Sincerely.

Reviewer 2 Report

The manuscript is much improved. The additional illustrations in the supplementary information are useful and it would be good to refer to these in the text.

Author Response

(The authors gave the same response as above.)
